# Remarkable Phenotypic Virulence Factors of *Microsporum canis* and Their Associated Genes: A Systematic Review

**DOI:** 10.3390/ijms25052533

**Published:** 2024-02-22

**Authors:** Tania Vite-Garín, Norma Angélica Estrada-Cruz, Rigoberto Hernández-Castro, Claudia Erika Fuentes-Venado, Paola Berenice Zarate-Segura, María Guadalupe Frías-De-León, Macario Martínez-Castillo, Erick Martínez-Herrera, Rodolfo Pinto-Almazán

**Affiliations:** 1Sección de Estudios de Posgrado e Investigación, Escuela Superior de Medicina, Instituto Politécnico Nacional, Plan de San Luis y Díaz Mirón s/n, Col. Casco de Santo Tomas, Alcaldía Miguel Hidalgo, Ciudad de México 11340, Mexico; tania.vite.garin@gmail.com (T.V.-G.); acilegnablue@live.com.mx (N.A.E.-C.); cefvenado@hotmail.com (C.E.F.-V.); mmartinezcas@ipn.mx (M.M.-C.); 2Facultad de Ciencias, Universidad Nacional Autónoma de México, Investigación Científica, C.U., Coyoacán, Ciudad de México 04510, Mexico; 3Departamento de Ecología de Agentes Patógenos, Hospital General “Dr. Manuel Gea González”, Ciudad de México 14080, Mexico; rigo37@gmail.com; 4Servicio de Medicina Física y Rehabilitación, Hospital General de Zona No 197, Texcoco 56108, Mexico; 5Laboratorio de Medicina Traslacional, Escuela Superior de Medicina, Instituto Politécnico Nacional, Ciudad de México 11340, Mexico; 6Unidad de Investigación Biomédica, Hospital Regional de Alta Especialidad de Ixtapaluca, Pueblo de Zoquiapan, Ixtapaluca 56530, Mexico; magpefrias@gmail.com; 7Fundación Vithas, Grupo Hospitalario Vithas, 28043 Madrid, Spain; 8Efficiency, Quality, and Costsin Health Services Research Group (EFISALUD), Galicia Sur Health Research Institute (IISGS), Servizo Galego de Saúde-Universidade de Vigo (UVIGO), 36213 Vigo, Spain

**Keywords:** *Microsporum canis*, extracellular virulence factors, phenotypic, intracellular virulence factors

## Abstract

*Microsporum canis* is a widely distributed dermatophyte, which is among the main etiological agents of dermatophytosis in humans and domestic animals. This fungus invades, colonizes and nourishes itself on the keratinized tissues of the host through various virulence factors. This review will bring together the known information about the mechanisms, enzymes and their associated genes relevant to the pathogenesis processes of the fungus and will provide an overview of those virulence factors that should be better studied to establish effective methods of prevention and control of the disease. Public databases using the MeSH terms “*Microsporum canis*”, “virulence factors” and each individual virulence factor were reviewed to enlist a series of articles, from where only original works in English and Spanish that included relevant information on the subject were selected. Out of the 147 articles obtained in the review, 46 were selected that reported virulence factors for *M. canis* in a period between 1988 and 2023. The rest of the articles were discarded because they did not contain information on the topic (67), some were written in different languages (3), and others were repeated in two or more databases (24) or were not original articles (7). The main virulence factors in *M. canis* are keratinases, fungilisins and subtilisins. However, less commonly reported are biofilms or dipeptidylpeptidases, among others, which have been little researched because they vary in expression or activity between strains and are not considered essential for the infection and survival of the fungus. Although it is known that they are truly involved in resistance, infection and metabolism, we recognize that their study could strengthen the knowledge of the pathogenesis of *M. canis* with the aim of achieving effective treatments, as well as the prevention and control of infection.

## 1. Introduction

Dermatophytes are a group of keratinolytic and keratinophilic fungi, etiological agents of zoonoses or anthropo-zoonotic transmissions that secrete a variety of enzymes during the adhesion, germination, invasion and penetration of host tissues, through which they are able to degrade components of the skin and extracellular matrix to obtain nutrients. The enzyme spectrum, as well as the duration and intensity of the activity, changes between species and is even variable between strains of the same species [1,2,3]. They are distributed in nine genera (*Trichophyton*, *Microsporum*, *Epidermophyton*, *Nannizzia*, *Arthroderma*, *Paraphyton*, *Lophophyton*, *Ctenomyces* and *Guarromyces*) according to their morphology, clinical form and molecular characteristics and grouped according to their main habitat into anthropophilic, geophilic and zoophilic, although in general they can be isolated from any of the three environments [4,5]. According to this new classification, some of those who belonged to one genus became part of another, such as *Microsporum gypseum*, which is now called *Nannizzia gypsea*, *Microsporum nanum*, which is now known as *Nannizzia nana*, etc. [6].

Among the most common dermatophytes are the three species of the genus *Microsporum*: *M. audouinii* and *M. ferrugineum*, both anthropophilic, and *M. canis*, which is considered anthropo-zoophilic; all of them are associated with dermatophytosis in domestic animals and tinea capitis, tinea corporis and occasionally onychomycosis in humans [2,6,7,8,9,10]. In particular, *M. canis* is considered to be of medical importance because it is capable of infecting pediatric or immunocompromised patients. It causes severe inflammatory responses (inflammatory type *tinea capitis* including Celsus’ Cherion, favus, *tinea barbae* and *tinea corporis*, the latter sometimes being an extension of HIV-AIDS-related *tinea capitis*) (Figure 1). In addition, this fungus has a wide worldwide distribution and is difficult to control. For infection to occur, hosts must come into contact with the microorganism through sick domestic animals, especially cats, dogs and horses, sick people or soil [5,7,11,12,13].

Despite the fact that, genetically, *M. canis* is the most heterogeneous species of the genus, with approximately 943 unique genes and 6000 orthologs shared with the rest of the dermatophytes [5], it has low intraspecific variability even between strains of different geographical origins or from different hosts, which makes its genotyping difficult. It is an anamorphic fungus that, in the laboratory, presents sexual compatibility with strains of the *Artrhoderma otae* complex; however, it is predominantly asexual, although the coexistence of clonal and recombinant populations in nature is not ruled out. It is also heterothallic, with its sexual compatibility factor determined by the *MAT* locus, presenting the idiomorph *MAT 1-1* in most of the strains analyzed to date [5,14,15,16].

A successful infection with *M. canis* is considered to be multifactorial and depends on characteristics such as genetics, maturity, immune system functioning, and the nutritional status of the host, as well as environmental conditions, mainly temperature and humidity [3]. This review will bring together the information known to date on the mechanisms, enzymes and their associated genes involved in the pathogenesis of *M. canis*, in addition to establishing an overview of those lines of research that need to be explored in detail for the establishment of effective methods for the prevention, control and treatment of the disease caused by this fungus.

## 2. Methods

Four databases, EBSCO (https://www.ebsco.com/es/productos/bases-de-datos [accessed on 10 December 2023]), SCOPUS (https://www.scopus.com/ [accessed on 10 December 2023]), SCIELO (https://scielo.org/es/ [accessed on 10 December 2023])) and PUBMED (https://www.ncbi.nlm.nih.gov/pmc/ [accessed on 10 December 2023]), were searched for publications under the MeSH terms “*Microsporum canis*”, “virulence factors”, “keratinases”, “metalloproteases”, “subtilisins”, “aminopeptidases”, “dipeptidylpeptidases”, “aspartyl proteases”, “hemolysins”, “catalases”, “ureases”, “serine hydrolases”, “biofilms” and “dermatophytomas”. After the automated search in the databases and the elimination of duplicates, two of the independent authors (T.V. g and N.A.E.-C.) performed the review of the articles from the title and abstract to select their eligibility. When conducting the systematic review, the PRISMA 2020 guidelines for systematic reviews and meta-analyses were used (Figure 2) [17], from which a total of 147 articles were obtained. Of these, 46 articles were included under the following criteria: Original articles, in English or Spanish, on virulence factors associated with *M. canis* dermatophytosis and 101 were excluded for different reasons, namely, 24 were repeated in one or more of the consulted databases, 3 articles were written in Portuguese, 7 papers were review articles and the remaining 67 did not contain information related to the subject of this work. For the analysis of the quality of risk of bias, it was carried out in duplicate (E.M.-H and R.P.-A.) using two tools, which serve to analyze the quality and risk of bias of qualitative systematic reviews: the JBI Critical Appraisal Checklist for Systematic Reviews and Research Syntheses, which indicates in the overall assessment that an article meets the quality needed to be published, and the Critical Appraisal Skills Program (CASP), which states that a study has a logical development that makes it feasible for publication.

## 3. Results 

Virulence factors and host receptivity provide *M. canis* with the ability to evade the immune response by infecting keratinized tissues, often persistently. In general, for the studied strains, the production of extracellular enzymes that favor their survival, which allows them to digest tissues to obtain nutrients that can be assimilated and thus adapt to their environment regardless of the state of the skin they infect, is reported (Figure 3) [7,18,19].

In this review, we found a number of virulence factors (Figure 2) that can be classified as extracellular and intracellular factors, which are described below.

### 3.1. Extracellular Virulence Factors

Extracellular virulence factors are those that allow a fungus to survive, nourish and colonize host tissues. Among these, we can certainly highlight the production of enzymes, particularly proteases and their activity, as being fundamental in the degradation of keratinized tissues through the hydrolysis of their components, as occurs in other dermatophytes [2]. The different enzymes that have been reported perform various actions on the host tissue; for example, in addition to the keratinolytic role, there are also some proteinases involved in the modulation of the immune system, the progression of infection and the penetration of the pathogen [20,21,22]. In this section, enzymes and mechanisms that allow the fungus to cope with the immune response of hosts and treatment with antifungal drugs are mentioned. The virulence factors reported for *M. canis* are outlined here.

#### 3.1.1. Keratinases

Among the enzymes produced by dermatophytes, keratinases are among the most important proteases in dermatophyte infections and are considered to be one of the most important virulence factors during tissue infection [1]. Regarding enzymatic activity, it has been reported in different studies that it does not always show differences depending on the origin or clinical manifestations of the strain; however, Ramos et al. [2] and Viani et al. [23] reported that there is greater productivity and/or activity in those that come from symptomatic cats compared to asymptomatic cats and that there are usually several enzymes that exhibit keratinolytic, elastinolytic, and collagenolytic activity. These enzymes solubilize keratin, and it has been observed in the laboratory that they are usually produced in high concentrations when the pH of the medium is around 7.5, in addition to having optimal temperatures between 35 and 50 °C [24]. Subsequently, for *M. canis*, Hamaguchi et al. [25] reported an enzyme with keratinase activity which they named Ecasa, which is formed by three subunits and has its homolog in other dermatophyte species such as *T. rubrum* and *T. mentagrophytes*. After the characterization of these, it was concluded that it is an enzyme between 45 and 32.5 kDa, tolerant to thermomycalases, with high concentrations of aspartic acid, glycine and alanine and being stable at 55 °C.

#### 3.1.2. Metalloproteases

Metalloproteases are endopeptidases grouped into 40 families according to the substrate they hydrolyze and require Zinc to perform their catalytic action [26]. For dermatophytes in general, the main proteases are deuterolisins, dipeptidylpeptidases, aminopeptidases, carboxylpeptidases, and serinendopeptidases. For *M. canis*, the most important are fungalisins, which, together with subtilisins, act on the epidermis of the hosts, allowing the adhesion of the fungus [12]. In particular, *M. canis* strains secrete, in vivo, at least three metalloproteinases (MEP1, MEP2 and MEP3) of the M36 family of fungalisins, molecules of approximately 43.5 kDa, which together with subtilases could reflect, evolutionarily, a certain degree of specialization in keratinized substrates.

Fungalisins allow the fungus to digest proteins in long-chain peptides due to their collagenolytic, elastinokeralytic and keratinolytic activity in the host, thus playing an important role in adhesion, nutrition, tissue invasion, and control of the host immune response [2,12,18,19,27,28,29,30].

#### 3.1.3. Subtilisins (Serine Endopeptidases)

Among the most important keratinolytic enzymes of *M. canis* are subtilisins [13,30,31]. For this fungus, Sub1 and Sub2 subtilisins have been reported as virulence factors due to their activity as homologous keratinases and the fact that they anchor to the host surface through the amino terminal end of the protein in conjunction with conformational changes [31]. Likewise, the Sub3 enzyme, characterized as a 31.5 kDa enzyme, is one of the most important and well-studied endoproteases among dermatophytes. It is necessary for tissue invasion due to its participation in adhesion, and it is expressed both in arthrochonidia and in hyphae in vitro, and in vivo, in cats and guinea pigs [31,32,33,34,35,36,37]. According to Baghut et al. [33] and Descamsp et al. [37], the three subtilisins are expressed in all dermatophytes; they are highly preserved and have major percentages of identity with proteases of *Trichophyton rubrum*. Specifically in *M. canis*, these subtilisins seem to be associated with delayed hypersensitivity reactions which may affect the susceptibility of the host to the pathogen and its persistence, leading to strains that produce chronic infections. The production of the enzyme Sub3 during *M. canis* infection indicates that the fungus is metabolically active. This enzyme is involved in adhesion and has been detected by immunohistochemical tests in hair follicles from biopsy samples from domestic cats, which researchers have assumed would support the creation of antibodies that would aim to decrease the production of this enzyme [31,38].

#### 3.1.4. Aminopeptidases

The leucin aminopeptidases Lap1 and Lap2 are enzymes homologous to carboxypeptidase A in the human pancreas. They are highly preserved among dermatophytes and produced in major concentrations in alkaline conformations. Its function is key for keratin degradation and N assimilation [39].

#### 3.1.5. Dipeptidylpeptidases

Vermount et al. [18,19] describe that the dipeptidyl peptidases that have been characterized for *M. canis* are part of the S9 family and are involved in tissue degradation and modulation of the immune response, contributing to fungal growth and virulence. In particular, the dipeptidyl peptidases DppIV and DppV have been reported, and it is suggested that these act in conjunction with aminopeptidases on the N-terminal end to obtain small peptides and favor nutrient production [30,39]. It is also suggested that these enzymes are involved in tissue colonization, participating in the degradation of collagen and elastin and in adhesion to tissues, thus facilitating this union by promoting the interaction between extracellular matrix components and the fungus, as has been observed in other microorganisms such as bacteria and protozoa [30].

#### 3.1.6. Aspartyl Proteases

Aspartyl proteases are enzymes commonly characterized in yeasts of the genus *Candida* and are associated with the degradation of component proteins of the host’s immune system. In the case of *M. canis*, the production of aspartyl proteases was observed both in vitro assays and ex vivo in cat skin analyses. However, its role in infection by this dermatophyte has not been explored [2].

#### 3.1.7. Hemolysins

Commonly, hemolysins that have been studied in pathogenic fungi act in the acquisition of iron; however, the role they play as a virulence factor in *M. canis* is uncertain, as some studies show low or no production and activity of this enzyme, which may be related to the little interaction of the fungus with the blood of the host [2,40,41]. On the other hand, there is some evidence of high hemolytic activity, cytotoxic effects in phagocytes, pore formation, cell lysis and balance between the host’s immune response and the pathogen’s tolerance to this response, especially in *M. canis* strains that come from symptomatic hosts (with skin lesions), facilitating deep tissue infection and participating in the acquisition of iron obtained from heme groups [1,7,42].

#### 3.1.8. Catalases

Catalases, in general, protect pathogenic fungi from reactive oxygen species that are produced as part of the host’s immune mechanisms. In the case of dermatophytes in general, they have been little studied and, in some research, such as that of Ramos et al. [2], it has been observed that catalase production and its enzymatic activity is low, which may be due to the fact that *M. canis* is an extracellular pathogen, so it would present low exposure to this type of compounds. However, a later study [7] noted that all strains produce catalases, with increased production and enzymatic activity in those hosts that show skin lesions, suggesting that these enzymes may be involved in the onset of infection.

#### 3.1.9. Ureases

According to Ramos et al. [2], the studies on ureases in dermatophytes that have been carried out until now have been qualitative analyses on the production of enzymes and have been mainly used for taxonomic purposes, since apparently not all strains produce them and the enzymatic activity also varies. It is suggested that the function of ureases during *M. canis* infection is to provide the fungus with a source of N for its growth.

#### 3.1.10. Serine Hydrolases

Zhang et al. [43,44] have acknowledged that among the proteases that may play an important role as virulence factors, *M. canis* produces ab-serine hydrolase, which functions as an esterase, acting through an active site formed by Ser/Hist/As, and that is believed to participate in various processes of the regulation of metabolism, with colony pigmentation and the growth and development of macroconidia.

#### 3.1.11. Biofilms and Dermatophytomas

The ability of various species of fungi to form structures that provide resistance against stressors or toxins in order to mitigate the effects of the host’s immune response is considered an important virulence factor, as it contributes to the process of colonization and infection of the host, may be difficult to treat or may even be removed by surgery [45,46,47,48,49]. Among these are biofilms, through which the fungus produces different types of molecules that help to establish itself on the tissues of the bone. For example, adhesins facilitate adhesion to biotic and abiotic surfaces, the overproduction of efflux pumps and extracellular matrix components, among others. In the particular case of *M. canis*, the biofilm is made up of a network of organized and cross-linked hyphae, connected by an extracellular matrix of polysaccharides. This physical barrier prevents the entry of antifungals, promotes the development of tolerant cells and decreases the interaction of the fungus with the host’s immune system [46,48,50,51,52,53,54]. Furthermore, the formation of subungual dermatophytomas has been described as thick-walled hyphae masses with conidia that limit the action of antifungals [49].

#### 3.1.12. Thermotolerance

According to Whang et al. [42], one of the reasons that dermatophytes do not usually cause deep infections in hosts is due to their optimal growth temperature, estimated to be around 25 °C, which is incompatible with the body temperature of the hosts. The elevation of temperature generates a decrease or increase in some metabolic processes, so thermotolerance would allow the strains that present it to infect at deeper levels than in non-invasive dermatophytes. According to their analyses, these authors report that invasive strains cultured at 37 °C showed the over-regulation of genes related to DNA replication, the repair of transcription errors, N-glycan biosynthesis and ribosome biosynthesis, while those that developed at 28 °C did not show this behavior.

Among the virulence factors related to thermotolerance, for pathogenic fungi, some heat shock proteins (HSPs) have been characterized, which are involved in the survival of fungi in stressful situations [47]. HSPs are highly preserved chaperone proteins, reported for commensal fungi such as *Candida* spp., which also chaperone to those that cause systemic mycoses in mammals and some dermatophytes, mainly *T. rubrum*, for which it is suggested they perform functions related to resistance to antifungals, adhesion during tissue invasion and the maintenance of various cellular processes. Additionally, HSPs direct the host’s immune response and modulate gene expression through signal transduction, raising the ability to grow in keratinized tissues and allowing fungi to adapt to stress conditions, particularly in invasive strains that trigger immunological reactions in the host, e.g., oxidative stress and the optimization of metabolic activity when the temperature increases to around 37 °C [47,55]. The role of HSPs in the infection caused by invasive strains of *M. canis* is currently unknown, but considering that they are highly preserved molecules, it would be expected that they become a good target for the development of treatments for those deep infections by this fungus.

Finally, it was found that some strains may be poor secretors of enzymes such as gelatinases, elastases, DNAses and lipases, which have been mainly linked to infections in human tissues with the presence of lesions. These act by allowing the invasion of the host through the hydrolysis of extracellular matrix components, particularly collagen and elastin, converting them into polypeptides, peptides and amino acids [1,41,56,57]. However, the reported results are inconclusive and scarce, so it is considered that they may not be determinant enzymes for the establishment of infection. However, more studies are required in this regard.

### 3.2. Intracellular Virulence Factors

The overexpression of different genes is associated with the infectious processes of *M. canis*, so it is suggested that they may actively participate as virulence factors, as they improve metabolic activity, nutrient acquisition and reproduction. The genes that have been considered virulence factors are mentioned in Table 1 [5,35,39,43,44].

## 4. Discussion

*M. canis* is a fungus of medical importance to both humans and pets. Due to its characteristics, *M. canis* is one of the main etiological agents of *tineas* and dermatophytoses worldwide [2,3,7,9,23,28,30,32,36,43,44,45]. The severity of the infection produced in some susceptible hosts and the difficulties that arise in establishing treatments or prevention strategies are the main causes of its dissemination [5,7,11,12,13]. The key to its success as a pathogen is mainly due to the expression of the virulence factors mentioned in the present review, which allow *M. canis* to nourish itself from keratinized tissues, adapt to different types of environments and evade the immune response of hosts through the synthesis of enzymes and the organization of their hyphae depending on the needs that the environment sets on them [1,2,3,5,7,11,12,13].

According to this review, the most studied virulence factors for *M. canis* are the different types of proteinases they are capable of producing, which are highly preserved among dermatophytes such as *T. rubrum* [4,12,20,21,22] and even share similarity with those secreted by other types of fungi, including opportunistic ones such as *Aspergilus* spp. [39,43].

In *M. canis*, the most common proteinases that play fundamental roles in host invasion are keratinases, fungalisins and subtilisins, which have different degrees of activity and levels of secretion. This will depend on the type of host (species, age, state of the immune system, etc.), the clinical manifestations (dry or inflammatory ringworms, whether asymptomatic or not, etc.) and the stage of the infection being analyzed (invasion, colonization or nutrition) [2,3,7,37].

Keratin is a protein that provides stability and protection to the epidermis of vertebrates and is one of the main defense strategies against the action of pathogens and other harmful factors for the individual [58]. These proteins are difficult to degrade, and therefore, dermatophytes require keratinases as their main proteinase, which are fundamental in the process of tissue invasion [1,2,23,24,25]. On the other hand, the main function of fungalisins and subtilisins is developed in the colonization process, allowing the adhesion of arthroconodia [2,12,18,19,26,27,28,29,30,31]. These are the three types of enzymes best studied in *M. canis* and in other dermatophytes, possibly because they are the ones that are secreted in the highest concentration and have the greatest activity on the components of the skin, hair and nails [31,32,33,34,35,36,37,38]. Nevertheless, this generates a bias in the information about the biological characteristics and pathogenesis of the fungus, hindering the development of prevention and control strategies. As these virulence factors are the most studied, it could be assumed that they are the only ones involved in pathogenesis.

From a different perspective, there are other factors that have been less studied, possibly because their synthesis and/or activity in dermatophytosis caused by *M. canis* is less common, or since dermatophytosis is not considered a deep mycosis, it does not require other virulence factors. Examples of these are thermotolerance, dermatophytomas, biofilms, and enzymes such as dipeptidylpeptidases, which are relevant for invasion and survival as mechanisms of resistance to the host’s immune system and antifungals [1,12,41,42,45,46,47,48,49,50,51,52,53,54,55,56,57].

Based on the above, it is important to take into account that keratinocytes are only distributed on the surface of the skin (epidermis), while the dermis is mainly made up of collagen and elastin, both of which are involved in the integrity of the skin, giving it strength, flexibility and firmness [59,60]. Dipeptidylpeptidases, along with metalloproteases, are enzymes that hydrolyze collagen and elastin, allowing the fungus to establish on deeper layers of the host’s skin than previously believed [12,18,19,28,30,39]. Another important factor to mention that has been little studied is the sulfite efflux pump, which is key to eliminating the sulfur compounds that are cytotoxic and that are produced by the internal degradation of the components of the epidermis and dermis (Table 1) [5,35].

Regarding metabolism in *M. canis*, it has been observed that there are enzymes important for the breakdown of peptides into amino acids, the assimilation of N, cysteine synthesis, serine hydrolysis and for the respiratory chain (Table 1) [39,43,44].

Likewise, enzymes such as hemolysins, catalases or ureases, which are also involved in the mitigation of the immune response and whose functions for *M. canis* are inferred from what has been observed in other pathogens (dermatophytes and non-dermatophytes) [1,2,40], remain unknown in terms of their conditions of production and involvement in the process of infection. It is important to note that these enzymes depend on the needs of the fungus related to the environment in which it is developing.

For this reason, it is essential to design more complete studies that allow us to understand the genetic and molecular characteristics of the pathogen and its mechanisms of pathogenesis, as well as to identify patterns in the synthesis and secretion of enzymes with a view to addressing dermatophytosis caused by *M. canis* as a public health matter and achieving strategies for the prevention, treatment and control of the disease.

## 5. Conclusions

Because *M. canis* is a pathogenic fungus that affects a vulnerable sector of both human and domestic animal populations, it is relevant for public health reasons. The establishment of more complete studies to elucidate the function of virulence factors such as dipeptidylpeptidases, hemolysins, catalases, aminopeptidases or serine hydrolases, which have been little studied and have been shown to be important in pathogenesis, may strengthen our knowledge of the mechanisms of invasion, colonization and nutrition of this fungus from keratinized tissues, with the aim of achieving effective treatments, as well as the prevention and control of infection.

## 6. Limitations and Perspectives

The main limitation of both this work and the way in which the study of *M. canis* has been approached is the lack of information generated about proteins, genes and mechanisms considered as virulence factors in more prevalent dermatophytes. This, together with the fact that it is a superficial pathogen, means that little attention has been paid to those factors that could be of importance in the pathogenesis and dissemination of the fungus.

Since this work is a systematic review, only those virulence factors that have been widely studied for this fungus are included, such as metalloproteases or keratinases, among others. The study of other proteins or mechanisms that are known to be part of the mechanisms of pathogenesis of other dermatophytes is proposed as a line of research that can be addressed in the future.

## Figures and Tables

**Figure 1 ijms-25-02533-f001:**
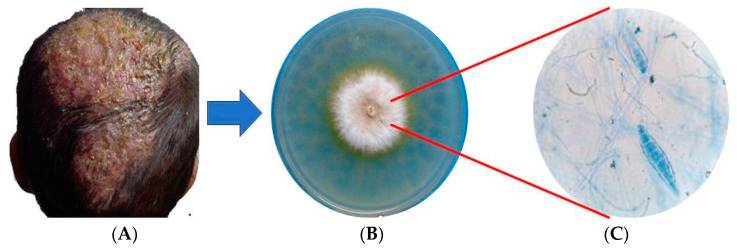
*Tinea capitis* caused by *Microsporum canis*. (**A**) Inflammatory *tinea capitis*, (**B**) Mycocel^®^ agar culture, (**C**) macroconidia stained with lactophenol blue.

**Figure 2 ijms-25-02533-f002:**
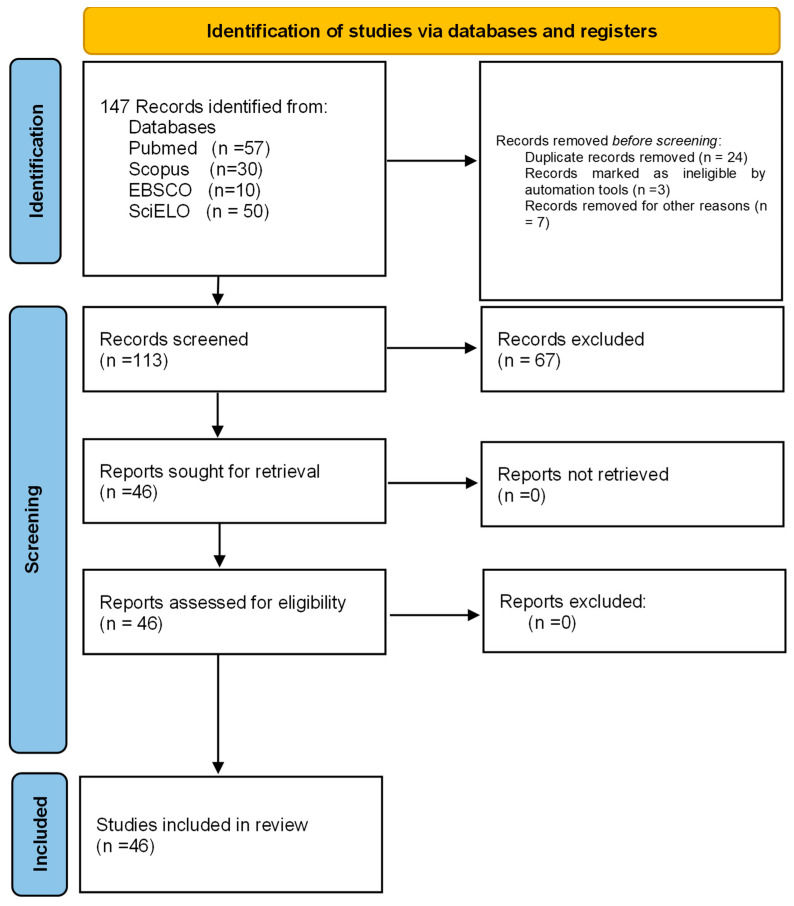
Prisma 2020 flowchart of the data extracted for the systematic review from bibliographic searches.

**Figure 3 ijms-25-02533-f003:**
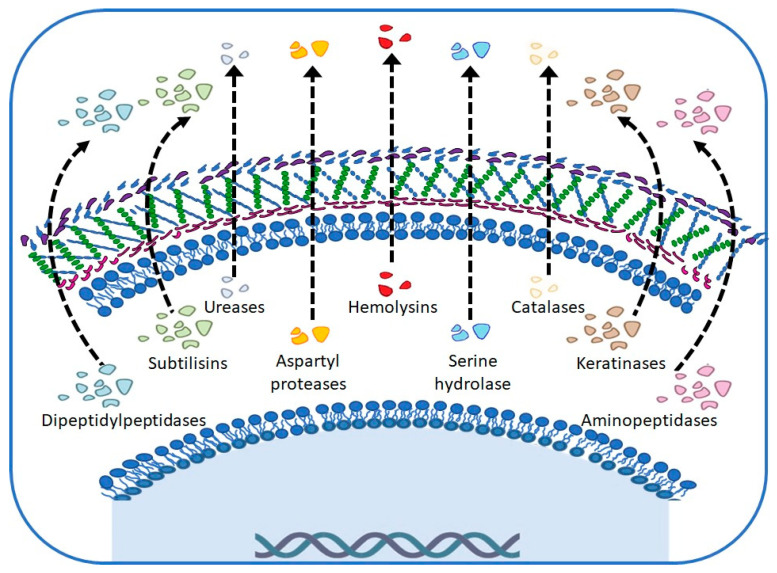
Main virulence factors in *Microsporum canis*.

**Table 1 ijms-25-02533-t001:** Genes considered as virulence factors due to their association with survival, nutrition and/or colonization of keratinized tissues by *M. canis*.

Gene	Protein	Function	Reference
*FSH1*	Serine hydrolase	Regulation of metabolism and production of macroconidia	[43,44]
*PQ-LRP*	Membrane protein	Cysteine synthesis	[43]
*NADH1*	Nicotine Adenine Dinucleotide	Respiratory chain	[43]
*P-GAL4*	Protein GAL4	Metabolism of N and production of conidia	[43]
*AreA*	Transcription factor GATA	N metabolism and regulation of proteolytic activity	[39]
*PacC*	pH signaling transcription factor	Regulation of proteolytic activity	[39]
*SSU1*	Sulfite efflux pump	Cytotoxic compound removal	[5,39]
*SUB1*	Subtilisin SUB1	Arthroconidia anchorage to the surface of the host	[12,31,36]
*SUB2*	Subtilisin SUB2	Anchoring or adhesion of the arthroconidia to the host’s skin	[12,31,36]
*SUB3*	Subtilisin SUB3	Arthrochonidia adhesion	[5,18,19,35,36,37]
*CDO1*	Cysteindioxygenase 1	Elimination of cytotoxic compounds	[35]
*MEP1-MEP5*	Metalloproteases 1 to 5	Collagenolytic, elastinokeralytic activity and keratinolytic	[12,28]
*DPPIV* and *DPPV*	Dipeptidylpeptidases IV and V	Polypeptide hydrolysis and degradation of collagen and elastin	[18,19,30,39]
*LAP1* and *LAP2*	Leucinaminopeptidases 1 and 2	Converting long-chain peptides to amino acids and short-chain peptides	[39]

## Data Availability

Not applicable.

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
