# Peer review of "Remarkable Phenotypic Virulence Factors of Microsporum canis and Their Associated Genes: A Systematic Review"

_ijms, 2024, doi:10.3390/ijms25052533_

Round 1

Reviewer 1 Report

Comments and Suggestions for Authors

A very interesting review article. However, I have a few comments:

1. First, the first sentence of the abstract, line 28. The authors wrote: "Microsporum canis is a yeast of worldwide....". This is a substantive error. Microsporum canis is not a yeast, it is a dermatophyte. Dermatophytes belong to filamentous fungi. They are not yeast. This absolutely needs to be corrected.

2. The manuscript is a systematic review type and the authors used the PRISMA scheme. This is the correct approach, but there is no risk of bias determination. In my opinion, the risk of bias analysis should be supplemented.

3. And a small note: genus Latin names should be written in Italic font. Please correct the genus name "Candida" in lines 206 and 270 and check the entire manuscript again for this purpose.

After making corrections and supplementing the risk of bias, the manuscript will present a level appropriate for publication.

Author Response

January, 2024

Subject: Revision and resubmission of manuscript ijms-2815918

Ms. Hui Ling Chia

Guest Editor

Special Issue “Antimicrobial, Anti-biofilm and Anti-virulence Activities of Novel Compounds”

Dear Dr. Mara Di Giulio 

We greatly appreciate the revisions made to our manuscript entitled “Systematic review of Microsporum canis remarkable phenotypic virulence factors” by Vite et al. We addressed the comments and suggestions from both reviewers. These revisions have been of great help to round out and improve our work. We included the reviewer's major comments immediately after this letter, responding to the suggestions.

All the authors appreciate very much the opportunity to correct the manuscript, hoping it is suitable for acceptance and publication in your prestigious journal with these changes.

Looking forward to hearing from you shortly, best wishes

Sincerely,

Erick Martínez-Herrera, PhD.

Sección de Estudios de posgrado,

Escuela superior de medicina, Instituto Politécnico nacional

Tel: +5557296300

Email: erickmartinez_69@hotmail.com

Reviewer’s comments, author responses, and manuscript changes

The referee’s comments are in italics, the author's responses are in blue

Answers to Reviewer 1 concerns:

A very interesting review article. However, I have a few comments:

  1. First, the first sentence of the abstract, line 28. The authors wrote: "Microsporum canis is a yeast of worldwide....". This is a substantive error. Microsporum canis is not a yeast, it is a dermatophyte. Dermatophytes belong to filamentous fungi. They are not yeast. This absolutely needs to be corrected.

Answer: Thank you for your valuable observation. We change the sentence to “Microsporum canis is a dermatophyte of worldwide…”

  1. The manuscript is a systematic review type and the authors used the PRISMA scheme. This is the correct approach, but there is no risk of bias determination. In my opinion, the risk of bias analysis should be supplemented.

Answer: Thank you for your valuable observation. We used two risk of bias analyzes: 1 JBI Critical Appraisal Checklist for Systematic Reviews and Research Syntheses; 2. Critical Appraisal Skills Programme (CASP); which are tools to analyze the quality and risk of bias of qualitative systematic reviews.

  1. And a small note: genus Latin names should be written in Italic font. Please correct the genus name "Candida" in lines 206 and 270 and check the entire manuscript again for this purpose

We appreciate the important comments made to the paper; the name of the genus Candida was placed in italics on the corresponding lines.

Answers to Reviewer 2 concerns:

The authors performed a systematic review of the virulence factors of Microscoporum canis. Comments:

  1. The title of the article mentions the “phenotypic virulence factors”, but the abstract mentions the genes relevant to the pathogenesis too. Please update the title to match the content of the article.

Answer: Thank you for your valuable suggestion. We change “Systematic review of Microsporum canis remarkable phenotypic virulence factors” to “Microsporum canis remarkable phenotypic virulence factors and their associated genes: A systematic review”.

  1. Please explicitly state the objective/question of the review. Is it to identify all reported virulence factors of  canis? In the abstract, the review aims to describe “mechanisms”, “enzymes” and “genes”, while in the Introduction section, the aim involves “mechanisms” and “molecules”.

Answer: Thank you for your valuable observation. The objective was unified in both the summary and the introduction as follows:This review will bring together the information known to date on the mechanisms, enzymes and their associated genes, involved in the pathogenesis of M. canis”

  1. What are the keywords used for the search? In the abstract, the following terms are mentioned "Microsporum canis", "virulence factors" and „each individual virulence factor”. In the „Methods” section only „Microsporum canis” and „virulence factors” are mentioned. Were terms for „each individual virulence factor” used? If yes, please mention them.

Answer: We agree with your observation, the error was corrected and all terms for each virulence factor were used in the materials and methods section as follows: “were searched for publications under the terms MESH Microsporum canis”, “virulence factors”, “keratinases”, “metalloproteases”, “subtilisins”, “aminopeptidases”, “dipeptidylpeptidases”, “aspartyl proteases”, “hemolysins”, “catalases”, “ureases”, “serine hydrolases”, “biofilms” and “dermatophytomas”

  1. Please clearly mention the inclusion and the exclusion criteria.

Answer: Thank you for your valuable observation. “Were included under the following criteria: Original articles, in English or Spanish, on virulence factors associated with M. canis dermatophytosis and 101 were excluded for different reasons, namely, 24 were repeated in one or more of the consulted databases, 3 articles were written in Portuguese, 7 papers were review articles and the remaining 67 did not contain information related to the subject of this work.”

  1. What kind of studies were included? 

Answer: Only original articles were used and letters to the editor, clinical cases, systematic reviews, reviews, multicenter studies, etc. were discarded.

  1. The inclusion of articles was made by two researchers or by just one?

Answer: The inclusion of the articles was carried out by more than two researchers

  1. Was the risk of bias assessed? If yes, by which methods?

Answer: Thank you for your valuable observation. We used two risk of bias analyzes: 1 JBI Critical Appraisal Checklist for Systematic Reviews and Research Syntheses; 2. Critical Appraisal Skills Programme (CASP); which are tools to analyze the quality and risk of bias of qualitative systematic reviews.

  1. What are the limitations of the study?

Answer: We agree with your observation. The limitations and perspectives of study were: The main limitation of both this work and the way in which the study of M. canis has been approached is the lack of information generated about proteins, genes and mechanisms considered as virulence factors in more prevalent dermatophytes. This, together with the fact that it is a superficial pathogen, little attention has been paid to those factors that could be of importance in the pathogenesis and dissemination of the fungus.

Since this work is a systematic review, only those virulence factors that have been widely studied for this fungus are included, such as metalloproteases or keratinases, among others. The study of other proteins or mechanisms that are known to be part of the mechanisms of pathogenesis of other dermatophytes is proposed as a line of research that can be addressed in the future.

  1. Lines 321-322 states that the most important virulence factors for M. canis are the proteinases. Why are the proteinases the most important ones?

Answer: We agree with your observation. In the text it was changed from “the most important virulence factors for M. canis are the proteinases” to “the most studied virulence factors virulence factors for M. canis”.

Reviewer 2 Report

Comments and Suggestions for Authors

The authors performed a systematic review of the virulence factors of Microscoporum canis.

Comments:

1.      The title of the article mentions the “phenotypic virulence factors”, but the abstract mentions the genes relevant to the pathogenesis too. Please update the title to match the content of the article

2.      Please explicitly state the objective/question of the review. Is it to identify all reported virulence factors of M. canis? In the abstract, the review aims to describe “mechanisms”, “enzymes” and “genes”, while in the Introduction section, the aim involves “mechanisms” and “molecules”.

3.      What are the keywords used for the search? In the abstract, the following terms are mentioned "Microsporum canis", "virulence factors" and „each individual virulence factor”. In the „Methods” section only „Microsporum canis” and „virulence factors” are mentioned. Were terms for „each individual virulence factor” used? If yes, please mention them.

4.      Please clearly mention the inclusion and the exclusion criteria.

5.      What kind of studies were included?  

6.      The inclusion of articles was made by two researchers or by just one?

7.      Was the risk of bias assessed? If yes, by which methods?

8.      What are the limitations of the study?

9.    Lines 321-322 states that the most important virulence factors for M. canis are the proteinases. Why are the proteinases the most important ones?

Comments on the Quality of English Language

Line 270: please italize “Candida spp.”

Author Response

(The authors gave the same response as above.)

Round 2

Reviewer 1 Report

Comments and Suggestions for Authors

A very interesting manuscript. The authors have made all suggested changes.

Author Response

February, 2024

Subject: Revision and resubmission of manuscript ijms-2815918

Dr. Mara Di Giulio 

Guest Editor

Special Issue “Antimicrobial, Anti-biofilm and Anti-virulence Activities of Novel Compounds”

Dear Dr. Mara Di Giulio 

We greatly appreciate the revisions made to our manuscript entitled “Systematic review of Microsporum canis remarkable phenotypic virulence factors” by Vite et al. We addressed the comments and suggestions from both reviewers. These revisions have been of great help to round out and improve our work. We included the reviewer's major comments immediately after this letter, responding to the suggestions.

All the authors appreciate very much the opportunity to correct the manuscript, hoping it is suitable for acceptance and publication in your prestigious journal with these changes.

Looking forward to hearing from you shortly, best wishes

Sincerely,

Erick Martínez-Herrera, PhD.

Sección de Estudios de Posgrado e Investigación

Escuela superior de medicina, Instituto Politécnico nacional

Tel: +5557296300

Email: erickmartinez_69@hotmail.com

Rodolfo Pinto Almazán

Sección de Estudios de Posgrado e Investigación,

Escuela superior de medicina, Instituto Politécnico nacional

Tel: +5557296300

Email: rodolfopintoalmazan@gmail.com

Reviewer’s comments, author responses, and manuscript changes

The referee’s comments are in italics, the author's responses are in blue

Answers to Reviewer 1 concerns:

A very interesting manuscript. The authors have made all suggested changes.

Answer: Thank you for your valuable comments

Answers to Reviewer 2 concerns:

The manuscript improved compared with the last version.

  1. Lines 28-30: Ambiguous sentence. Please rephrase to increase the clarity.

Answer: Thank you for your valuable suggestion. We change “Microsporum canis is a dermatophyte of worldwide prevalence as a causative agent of dermatophytosis in domestic animals and tineas in humans that requires various virulence factors to invade, colonize and nourish keratinized tissues.” to “Microsporum canis is a widely distributed dermatophyte, which is among the main etiological agents of dermatophytosis in humans and domestic animals. This fungus invades, colonizes and nourishes on the keratinized tissues of the host through various virulence factors”.

Introduction

  1. Although not forbidden (to my knowledge), it is unusual to use figures in the Introduction section. If possible, it can be moved.

Answer: Thank you for your valuable observation. We consider that figure No. 1 should be placed in the Introduction, since it is a reference figure on tinea capitis and M. canis as the main causal agent. ¨

Methods:

  1. I do have some concerns regarding the methodology. For example, at a simple PubMed search using “Microsporum canis” (https://pubmed.ncbi.nlm.nih.gov/?term=Microsporum+canis&filter=pubt.clinicaltrial&filter=pubt.meta-analysis&filter=pubt.randomizedcontrolledtrial&sort=date&sort_order=asc) there are 45 results. According to the PRISMA flowchart, there were 57 results. The MeSH term “Microscopum canis” provided 4 results https://www.ncbi.nlm.nih.gov/mesh/?term=Microsporum+canis.

In comment 1 of the methods, the reviewer comments that "at a simple PubMed search using "Microsporum canis"... and provides a link, in which it is observed that filters by type of work were included (for example: clinical trial, meta-analysis, etc.), however, this review was performed without including any filter by type of research, so the results show all the works in which virulence factors for M. canis are mentioned, of which only the original articles were used because it was a systematic review.

The systematic review was performed using as MeSH terms "Microsporum canis" and "virulence factors", so that by means of quotation marks they are considered as two terms and associated with each other limiting the results to only virulence factors reported for M. canis. This same process was performed for "Microsporum canis" associated to each individual virulence factor ("subtilisins", "biofilm"...). Duplicate articles from the searches we removed.

Finally, the 57 results for PubMed mentioned in PRISMA and in materials and methods include the original articles resulting from the searches "Microsporum canis" "virulence factors", "Microsporum canis" "Subtilisins", "Microsporum canis" "keratinases", etc. Some of the searches are as follows in Pubmed:

  1. https://pubmed.ncbi.nlm.nih.gov/?term=%22Microsporum+canis%22+%22virulence+factors%22
  2. https://pubmed.ncbi.nlm.nih.gov/?term=%22Keratinase%22+%22microsporum+canis%22
  3. https://pubmed.ncbi.nlm.nih.gov/?term=%22microsporum+canis%22+%22biofilm%22
  4. https://pubmed.ncbi.nlm.nih.gov/?term=%22Microsporum+canis%22+%22subtilisins%22

  1. Line 106 – the correct spelling is “MeSH” terms, not “terms “terms MESH”. Were MeSH terms searched or just keywords?

Answer: Thank you for your valuable observation. The error was corrected. “were searched for publications under the MeSH terms Microsporum canis”, “virulence factors”, “keratinases”, “metalloproteases”, “subtilisins”, “aminopeptidases”, “dipeptidylpeptidases”, “aspartyl proteases”, “hemolysins”, “catalases”, “ureases”, “serine hydrolases”, “biofilms” and “dermatophytomas”.  We search for MeSH terms

  1. Is the search strategy registered? If yes, please provide registration details. It is not mandatory, but it is advisable.

     Answer: We agree with your observation, although it is advisable, in this case we did not do it, as we considered that it was not necessary.

Results:

  1. In the Prisma flow chart: Reports sought for retrieval (46), but the number of reports not retrieved is still 46, while 46 reports were assessed for eligibility. This is clearly a mistake.

     Answer: Thank you for your valuable observation. The error was corrected, was written 0 in reports not retrieved

  1. The authors state that they performed the risk of bias assessment using two instruments, but the results of the assessment are not represented in the results. Please provide the evidence synthesis, otherwise, all that effort will be in vain.

Answer: We agree with your observation. The instruments for the risk of bias assessments used for this paper are among the most recommended for qualitative analyses, as in the case of this systematic review; however, they do not have a numerical rating. Information on the results of the assessments is included in the methods section and the instruments are appended in supplementary material.

The text included in methods is as follows

“For the analysis of the quality of risk of bias, it was carried out in duplicate (E.M.-H and R.P.-A.) using two tools, which serve to analyze the quality and risk of bias of qualitative systematic reviews: JBI Critical Appraisal Checklist for Systematic Reviews and Research Syntheses, which indicated in the overall assessment that the article meets the quality to be published, and Critical Appraisal Skills Program (CASP), which states that this study has a logical development that makes it feasible for publication”.

Reviewer 2 Report

Comments and Suggestions for Authors

The manuscript improved compared with the last version.

Abstract:

1.      Lines 28-30: Ambiguous sentence. Please rephrase to increase the clarity.

Introduction

2.      Although not forbidden (to my knowledge), it is unusual to use figures in the Introduction section. If possible, it can be moved.

Methods:

1.      I do have some concerns regarding the methodology. For example, at a simple PubMed search using “Microsporum canis” (https://pubmed.ncbi.nlm.nih.gov/?term=Microsporum+canis&filter=pubt.clinicaltrial&filter=pubt.meta-analysis&filter=pubt.randomizedcontrolledtrial&sort=date&sort_order=asc) there are 45 results. According to the PRISMA flowchart, there were 57 results. The MeSH term “Microscopum canis” provided 4 results  https://www.ncbi.nlm.nih.gov/mesh/?term=Microsporum+canis.

2.      Line 106 – the correct spelling is “MeSH” terms, not “terms “terms MESH”. Were MeSH terms searched or just keywords?

3.      Is the search strategy registered? If yes, please provide registration details. It is not mandatory, but it is advisable.

Results:

4.      In the Prisma flow chart: Reports sought for retrieval (46), but the number of reports not retrieved is still 46, while 46 reports were assessed for eligibility. This is clearly a mistake.

5.      The authors state that they performed the risk of bias assessment using two instruments, but the results of the assessment are not represented in the results. Please provide the evidence synthesis, otherwise, all that effort will be in vain.

Comments on the Quality of English Language

Minor editing of English language required

Author Response

(The authors gave the same response as above.)

Round 3

Reviewer 2 Report

Comments and Suggestions for Authors

The manuscript improved.

Comments on the Quality of English Language

The quality of English language is fine